# Polymorphism of the Myostatin (*MSTN*) Gene in Landes and Kielecka Geese Breeds

**DOI:** 10.3390/ani10010010

**Published:** 2019-12-19

**Authors:** Grzegorz Smołucha, Anna Kozubska-Sobocińska, Anna Koseniuk, Kacper Żukowski, Mirosław Lisowski, Bartosz Grajewski

**Affiliations:** 1Department of Animal Molecular Biology, National Research Institute of Animal Production, Krakowska 1, 32-020 Balice, Poland; anna.sobocinska@izoo.krakow.pl (A.K.-S.); anna.koseniuk@izoo.krakow.pl (A.K.); 2Department of Cattle Breeding, National Research Institute of Animal Production, Krakowska 1, 32-020 Balice, Poland; kacper.zukowski@izoo.krakow.pl; 3Department Reproductive Biotechnology and Cryoconservation, National Research Institute of Animal Production, Krakowska 1, 32-020 Balice, Poland; miroslaw.lisowski@izoo.krakow.pl; 4Waterfowl Genetic Resources Station in Dworzyska, Experimental Station in Kołuda Wielka, National Research Institute of Animal Production, 62-035 Kórnik, Poland; bartosz.grajewski@izoo.krakow.pl

**Keywords:** polymorphism, geese, kielecka, landes, MSTN, myostatin

## Abstract

**Simple Summary:**

Identification of mutations in the myostatin gene, affecting the occurrence of the double muscling phenotype in some breeds of beef cattle, was an impetus for further analysis and identification of mutations within this gene in other animal breeds, characterized by increased meat performance parameters. The number of geese in poultry livestock production in Poland is small. The native geese breeds can be successfully used to produce high-quality poultry meat and can be a very good source of goose meat production for regional and organic products. The aim of the study was to identify a mutation in the *MSTN* (Myostatin) gene and investigate whether this polymorphism can affect body weight in different periods of life in Landes and Kielecka breeds. Measurements of the examined trait were taken with time intervals to demonstrate the putative effect of the identified SNP (Single Nucleotide Polymorphism) on body weight over the course of bird growth. In conclusion, the identified c.1231C>T polymorphism suggests a possible link between the polymorphism and the BW (body weight) of Kielecka geese in the 12th week of life. The most significant factors affecting the BW values in geese are breed and sex.

**Abstract:**

Myostatin, also known as growth differentiation factor 8 (GDF8), belongs to the TGF-β superfamily of proteins. MSTN is a highly conserved protein that acts as a negative regulator of skeletal muscle growth. Loss of myostatin functionality causes the phenotype to appear in the form of ‘double musculature’, among others in cattle, sheep, and house mice. The presented results of the research were carried out on two geese breeds—Landes and Kielecka. The aim of the study was to identify mutations in the *MSTN* gene and study their impact on body weight in both geese breeds in different periods of life. Analysis of the obtained results showed the existence of polymorphism in exon 3 (c.1231C>T) and suggested a possible association (*p* < 0.05) between BW and genotype in 12 weeks of life in male Kielecka geese breed. The identified polymorphism may be one of the factors important for improving body weight in the studied Kielecka breed, therefore, it is necessary to conduct further research on a larger population of geese breeds in order to more accurately estimate the effect of the identified SNP c.1231C>T on BW in geese.

## 1. Introduction

Proteins belonging to the transforming growth factors beta superfamily (TGF-β) include three isoforms (TGF-β1, -β 2 and -β3), activin and inhibin, growth factors (GDF), bone morphogenetic proteins (BMPs), and anti-Mullerian hormone (AMH), which consists of more than 35 proteins [1]. These proteins play a significant role in embryogenesis in mammals, amphibians, and insects, as well as in bone development, wound healing, hematopoiesis, and in the body’s response to inflammation [2,3,4,5]. Together with growth factors—GDF9 and GDF9b (BMP15)—they also play a significant role in mammal reproductive processes [6,7,8,9]. Myostatin (MSTN), also known as growth and differentiation factor 8 (GDF8), belongs to the TGF-β superfamily of proteins. In pig, buffalo, cow, fish (zebrafish), birds, and domestic mouse, the myostatin gene consists of 3 exons separated from each other by 2 introns and encodes 376 amino acid precursor protein [10]. Myostatin is expressed not only in skeletal muscle but also in the heart and adipose cells of adult animals [11]. Myostatin acts as a negative regulator of muscle growth and is associated with the growth and differentiation of the skeletal muscles of animals [12,13,14,15,16]. The loss of functionality of the *MSTN* gene caused by mutation occurring in the STOP codon resulted in the double-muscled phenotype in some breeds of beef cattle [12,17] and sheep [13,14,15]. The mutations identified in the *MSTN* gene were also associated with race outcomes of dogs [16] and horses [18]. Xu et al. [19] identified three SNP polymorphisms in the *MSTN* gene but only two were associated with breast muscle traits in Pekin duck. The T129C SNP had a significant association with breast muscle thickness, and the second one T952C had a significant association with the fossilia ossis mastoid length. Zhao et al. [20] investigated the effect of Myostatin and Myogenin on Zi and Rhine geese skeletal muscle growth. The authors found a negative association between *MSTN* gene expression in breast muscle and body weight, breast muscle weight and breast muscle percentage. Recently, there has been a noticeably increased consumer interest in unprocessed, high-quality food. Recent studies proved the high value of lipids and amino acids composition of local geese breeds [21]. It is well accepted that the body weight, as well as qualitative traits in geese, are strongly dependent on genotype [22,23,24]. However, the molecular background of meat production remains to be elucidated. For this study, we choose two local goose breeds significantly different in their muscularity and body weight, as well as the high breeds diversity of these traits. Landes geese, present in Poland since 1986, constitute a phenotypically unified population distinguished by good health, good reproductive traits, and meat efficiency [25]. The second breed, Kielecka goose, is native to southern Poland, characterized by very good muscularity, low fatness but poor meat production [26]. Both breeds have been included in the genetic resources program [27].

The aim of the study was to find and investigate whether the mutation in the *MSTN* gene affects body weight in different periods of life in Landes and Kielecka breeds.

## 2. Materials and Methods

### 2.1. Animals

For the study, 2 geese breeds (246 geese: Kielecka (109:82 Female, 27 Male) and Landes (137:100 Female, 37 Male), significantly different in their muscularity and body weight (BW) parameters as well as the high breeds diversity of this trait were selected for the study. The geese were kept in the same environmental conditions. The birds were fed according to the breeding goose nutrition program (Appendix A).

### 2.2. Measurements of Body Weight

The geese were weighed in WGRS NRIAP (Waterfowl Genetic Resources Station-National Research Institute of Animal Production). Measurements of body weight were made in the 8th, 12th and 95th weeks of life. Birds were weighed on a suspended Axis weight (max. 15 kg, min, 100 g). Birds were put into the scale (with their upper legs) thus that the movement of the bird was prevented—the wings were immobilized. Such a position during the moment of weighing caused relatively fast stabilization of the weight.

### 2.3. Molecular Analysis

DNA isolation was performed using the Sherlock AX reagent kit (A&A Biotechnology, Gdynia, Poland), according to the protocols provided by the manufacturer. The DNA material was isolated from the feathers, thus there was no need for the consent of the bioethics committee. 

### 2.4. Polymerase Chain Reaction (PCR)

Amplification and sequencing of the 3 myostatin gene exons were performed using the primers shown in Table 1. The PCR reaction was performed in 25 μL volume containing: 12 µL of PCR-grade water, 2.5 µL of PCR buffer with 15 mM MgCl2 (QIAGEN, Hilden, Germany), 5 µL of Q-Solution (5x; QIAGEN), 3 µL of 10 mM dNTPs (APPLIED BIOSYSTEMS, Foster City, CA, USA), 0.25 µL of primer mix (each 100 pmol/µL) (Table 1), 0.25 µL of HotStarTaq DNA polymerase (5 U/µL QIAGEN) and 2 µL of DNA isolate (50ng/ µL). The PCR thermal program was as follows: 15 min of initial activation step at 95 °C (Polymerase is activated by this heating step), 35 cycles of denaturation at 95 °C for 30 s, annealing at 61 °C for 59 s and primer extension at 72 °C for 120 s. The final extension was conducted at 72 °C for 10 min.

### 2.5. Sequencing Method

The PCR product was purified using the EXOSAP—it enzyme (Affymetrix, Santa Clara, CA, USA) and sequenced from both complementary strands using the BigDye Terminator v3.1 sequencing kit (Thermo Fisher Scientific, Waltham, MA, USA) and primers used for PCR reaction separately for forward and reverse primers. The reading of the sequencing reactions was carried out in the 3130xl Genetic Analyzer capillary sequencer (Applied Biosystems, Foster City, CA, USA). 

### 2.6. Alignment

The obtained sequences between individuals of the same breed as well as between breeds were aligned with the goose reference sequence (GenBank No. XM_013178647). We also did the alignment of the *MSTN* 3 exon for a consensus sequence of the Kielecka and Landes, as well as the goose reference sequence. The BioEdit Sequence Alignment Editor program was used [28].

### 2.7. Statistical Analyses

Associations between the analyzed features and the examined SNP polymorphism were estimated using the GLM model, using the Duncan multiple range tests. The differences within, between breeds, sexes, and genotypes were estimated according to the statistical model:Y*_ijkl_* = *a_i_* + *b_j_* + *c_k_*(1)
where *a_i_* = breed (*i* = Kielecka, Landes), *b_j_* = sex (*j* = male, female), *c_k_* = genotype (*k* = CC, CT, TT).

The Hardy–Weinberg equilibrium was tested using the Court Lab-HW calculator (Michael H. Court (2005–2008)).

### 2.8. Phylogenetic Analyses

To assess genetic distance of analyzed breeds and other bird species, sequences of *MSTN* gene of Kielecka and Landes breed as well sequences from GeneBank database for duck (Anas platyrhynchos, GeneBank Accession No XM_005011412.4), chicken (Gallus gallus, GeneBank Accession No NM_001001461.1), and pigeon (Columbia Livia, GeneBank Accession No NM_001282809.1) were used. The MEGA Molecular Evolutionary Genetics A software (MEGA7: Molecular Evolutionary Genetics Analysis version 7.0 for bigger datasets) was used to calculate genetic distance based on Tamura–Nei analysis model and construct maximum likelihood phylogenetic tree [29].

## 3. Results

Analysis of the obtained sequences allowed us to identify one mutation in both geese breeds. The SNP c.1231C>T (Figure 1) is a synonymous mutation in 3′ UTR (untranslated region) of the *MSTN* gene. The mutation depicted in Figure 2 presents the alignment of the *MSTN* 3 exon fragment of the consensus Kielecka/Landes sequence with the reference sequence (GenBank No. XM_013178647). 

In Table 2, allele frequencies and genotypes are shown. Both populations are in Hardy–Weinberg equilibrium (HWE) (*p* < 0.05). 

The phylogenetic analysis of a fragment of the *MSTN* exon3 of two goose breeds studied, and the sequences of duck, chicken, and pigeon are depicted in Figure 3. The closest evolutionary relationship was revealed for goose and duck and for these both species common ancestors are chicken and pigeon.

### Association Analysis of SNP with Body Weight

The results for BW measurement for each breed, sex, and genotype are shown in Table 3 for the age of 8 weeks, in Table 4 for the age of 12 weeks, and in Table 5 for the age of 95 weeks. The association analyses between the *MSTN* gene polymorphism and body weight trait for the Landes and Kielecka geese are shown in Figure 4, Figure 5 and Figure 6. All studied groups were analyzed by age, breed, sex, and genotype.

In the 8 weeks of life, the male Landes and Kielecka geese were heavier than females of each breed (*p* < 0.05) and females of Landes were of similar weight as males of Kielecka (*p* < 0.05) (Table 3, Figure 4). However, there was no association between genotypes and each BW measurement. 

At 12 weeks of age, the highest BW measurements were found in Landes males whereas the lowest BW values were noted for Kielecka females. These two groups were significantly different (*p* < 0.05). For both groups, there was no association between the BW results and the *MSTN* genotype. The CC homozygotes of Kielecka males and Landes females of all genotypes were of similar weight. The Kielecka male geese with CC genotypes were significantly heavier than individuals with CT and TT genotypes (*p* < 0.05) (Figure 5). In Table 4, the results of weight measurements with division into breeds and genotypes are shown. 

As was depicted in Figure 6, in the 95th week, statistically significant differences in body weight between the breeds were observed (*p* < 0.05). In the Landes breed, the greatest differences were found between females with CT genotype and males with CC genotype (*p* < 0.05). Moreover, in the Kielecka breed, there was no statistically significant difference between males and females. In Table 5, the results of body weight in both geese breeds were shown. 

## 4. Discussion

The participation of geese in poultry livestock production in Poland is small. This structure is similar to that observed in the European Union and is mainly caused by the stabilization of meat consumption of particular bird species. In Poland, goose meat production is mainly carried out with the use of W-31 hybrids, derived from the breeding set ♂ W-33 × ♀ W-11, i.e., based on the highly efficient Kołudzka white goose [30]. Recently, there has been a noticeably increased consumer interest in unprocessed, high-quality food. This trend is reflected in food products from small farms, the production of which is often based on the use of breeds with lower yield but characterized by increased resistance on diseases and environmental conditions and good use of feed.

The identification of mutations in the myostatin gene affecting the occurrence of the double-muscled phenotype in some breeds of beef cattle was the impetus for further analysis and identification of mutations within this gene in other animal breeds characterized by increased parameters of meat performance. In the case of cattle, the presence of deletion of 11 nucleotides in the 3 exons of the MSTN gene in homozygous form resulted in muscle hypertrophy presence in Belgian Blue cattle [12]. In sheep of different breeds, the c.1232G>A polymorphism in the 3 ‘UTR region was identified causing a change in the amino acid sequence to the stop codon, resulting in decreased myostatin production and, consequently, increased muscularity in sheep [13,14,15]. In dogs, the 939-940delTG deletion results in a change of cysteine to the stop codon. Dogs with this mutation are characterized by better results in races [16]. In addition, in the case of racehorses, the effect of g.66493737C>T mutation in MSTN on the results of races at different distances was demonstrated [18]. In pigs, 15 polymorphic sites have been identified, and these mutations are associated with the weight of piglets at birth and daily BW gains during the 60–100th day of fattening [31].

In the case of poultry, mutations in the *MSTN* gene were identified in chickens selected for meatiness (3556T>C; 3581T>A) and egg-laying performance (3360T>C; 3412A>G; 3533A>G; 3624A>T; 3656A>G) [32]. Zhang et al. [33] suggested that the mutation G2283A, detected in *MSTN* exon 1, has the potential as a genetic marker for body weight traits in the Bian chicken. Five polymorphisms at the 5′ and 3′ promoter sites of the *MSTN* gene were shown to influence bone growth and fat metabolism [34]. In ducks of Pekin breed, the 129T>C polymorphism was associated with the thickness of the pectoral muscle [19]. In studies carried out by Zhong [20], 6 polymorphisms were identified in the 3 exons of the *MSTN* gene in Sansui ducks, of which g.106>A was associated with slaughter traits.

In our research, no polymorphism was identified in exon 1 or 2 of the *MSTN* gene. We found, however, a polymorphism c.1231C>T in the exon 3 of the *MSTN* gene. The SNP polymorphism was localized 3 bp below CDS (coding DNA sequence) in the 3′ UTR region, which is responsible for translation control [35]. 

Statistical analyses showed significant differences (*p* ≤ 0.05) between breeds and between sexes within breeds for the studied trait. The obtained results demonstrate the high impact of breed and sex on body weight in the studied geese breeds, which is consistent with studies conducted elsewhere [22,23]. Based on the analysis of genotypes and the occurrence of significant statistical relationship for the characteristics of BW, the identified polymorphism cannot be clearly linked to the body weight of the studied geese breeds. However, it is believed that conducting further research on a larger population of different geese breeds could estimate more accurately the effect of the identified SNP c.1231C>T on body weight in geese.

## 5. Conclusions

In conclusion, the identified c.1231C>T polymorphism suggests a possible link between the body weight of Kielecka geese in the 12th week of life. The most significant factors affecting the BW values in geese are breed and sex.

## Figures and Tables

**Figure 1 animals-10-00010-f001:**
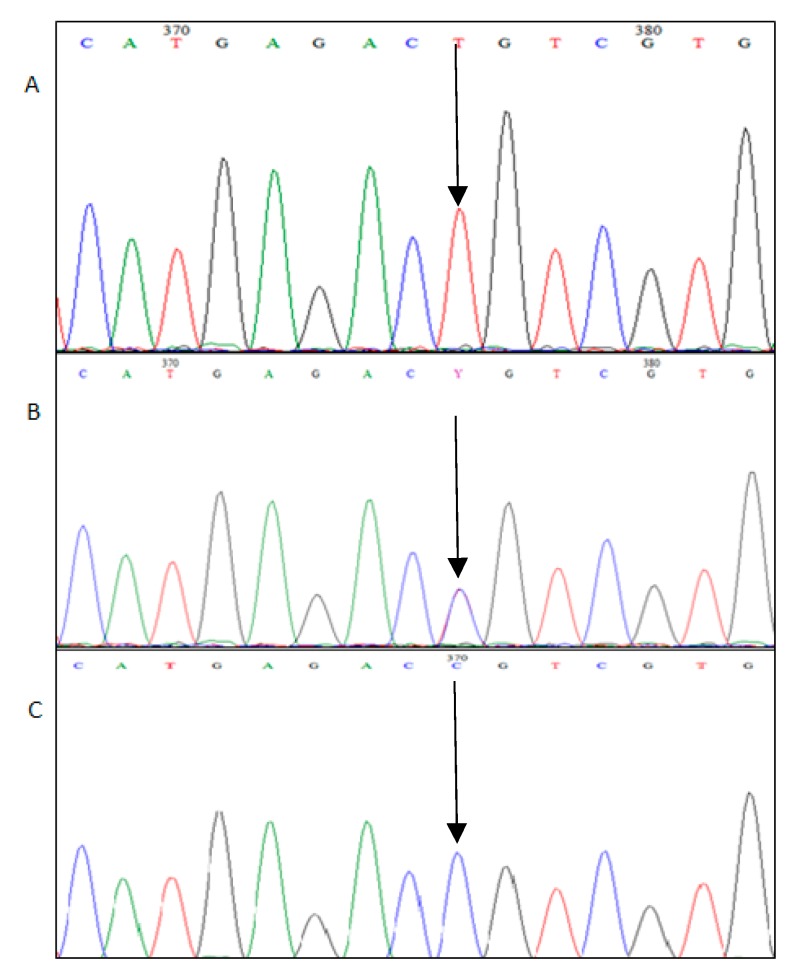
Chromatogram fragment of the Myostatin (*MSTN*) exon3 containing-c.1231C → T polymorphism. (A) Homozygote TT, (B) heterozygote C/T, (C) homozygote CC.

**Figure 2 animals-10-00010-f002:**
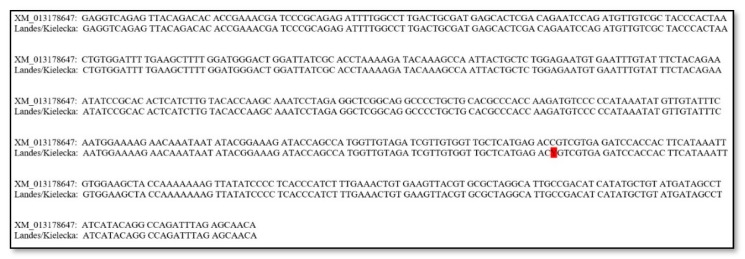
Alignment of the fragment of the *MSTN* gene with SNP c.1231 C>T marked in red.

**Figure 3 animals-10-00010-f003:**
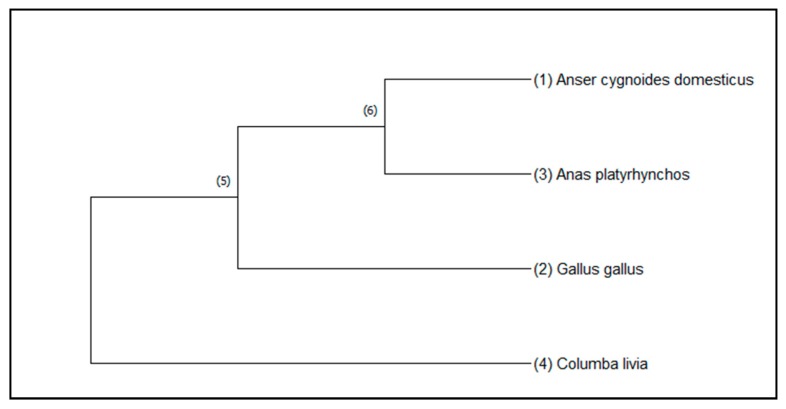
Unrooted maximum likelihood tree constructed from the Tamura–Nei model of the MSTN sequences of Kielecka and Landes breed (Anser cygnoides domesticus), Anas platyrhynchos (duck), Gallus gallus (chicken), Columbia Livia (pigeon).; bootstraps of 1000 replicates.

**Figure 4 animals-10-00010-f004:**
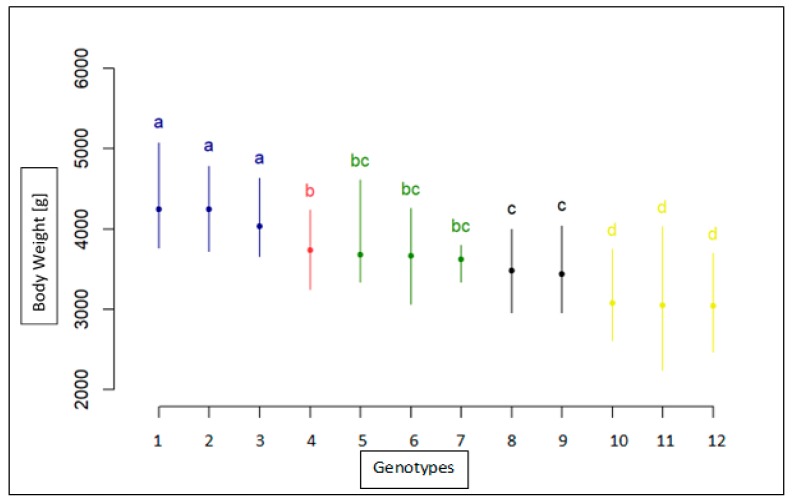
Statistical characteristics of association of c.1231C>T with body weight in 8-week old geese both breeds(1-TT(Male) Landes, 2-CT(Male)Landes, 3-CC(Male)Landes, 4-CT(Female)Landes, 5-CC(Female)Landes, 6-TT(Female)Landes, 7-CC(Male)Kielecka, 8-CT(Male)Kielecka, 9-TT(Male)Kielecka, 10-TT(Female)Kielecka, 11-CC(Female)Kielecka, 12-CT(Female)Kielecka). Genotypes with different letters show significant differences between groups (a, b, c, d: *p* ≤0.05).

**Figure 5 animals-10-00010-f005:**
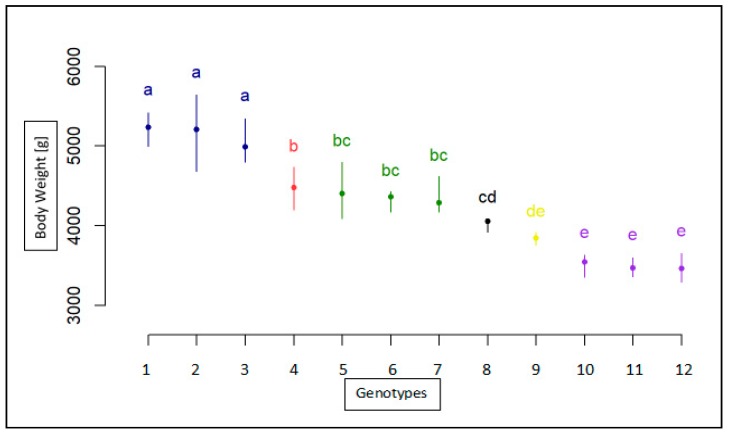
Statistical characteristics of association of c.1231C>T with body weight in 12-week old geese both breeds(1-CT(Male) Landes, 2-TT(Male)Landes, 3-CC(Male)Landes, 4-CC(Male)Kielecka, 5-TT(Female)Landes, 6-CC(Female)Landes, 7-CT(Female)Landes, 8-CT(Male)Kielecka, 9-TT(Male)Kielecka, 10-CC(Female)Kielecka, 11-TT(Female)Kielecka, 12-CT(Female)Kielecka). Genotypes with different letters show significant differences between groups (a, b, c, d, e: *p* ≤0.05).

**Figure 6 animals-10-00010-f006:**
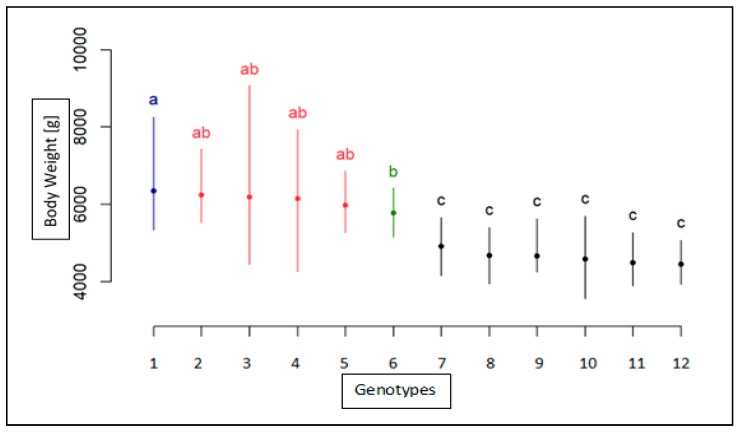
Statistical characteristics of association of c.1231C>T with body weight in 95-week old geese both breeds (1-CC(Female) Kielecka, 2-CC(Female)Landes, 3-CC(Male)Kielecka, 4-CC(Male)Landes, 5-CT(Female)Kielecka, 6-CT(Female)Landes, 7-CT(Male)Kielecka, 8-CT(Male)Landes, 9-TT(Female)Kielecka, 10-TT(Female)Landes, 11-TT(Male)Kielecka, 12-TT(Male)Landes). Genotypes with different letters show significant differences between groups (a, b, c: *p* ≤0.05).

**Table 1 animals-10-00010-t001:** Primers used for the amplification and sequencing of the myostatin gene of the Landes and Kielecka geese.

Primer Name	Primer Sequence		Tm * (°C)	Product Length (bp)
MSTNex3 F	Exon 3	5′ TTCCGGTTCCTTTTCCTCTT 3′	61	577
MSTNex3 R		5′ TCTGCAGCTTGTGTTGCTCT 3′		
MSTNex1 F	Exon 1	5′ TCAGATTGCATTTGCTTTCA 3′	61	497
MSTNex1 R		5′ AGACGAAAGCAGCAGGGTTA 3′		
MSTNex2 F	Exon 2	5′ TTTTTGTTCCCTGTTCAGTAATC 3′	61	473
MSTNex2 R		5′ TGCTTTCCAATAAAATGCAAGA 3′		

* PCR annealing temperature.

**Table 2 animals-10-00010-t002:** Frequency of c.1231 C>T alleles and genotypes in the *MSTN* gene in the Landes and Kielecka breeds and p-values for deviation from the Hardy–Weinberg equilibrium.

Breed	Genotypes			Alleles		HWE *p*-Value
	CC	CT	TT	C	T	
**Landes**	0.292	0.496	0.212	0.54	0.46	0.39
**Kielecka**	0.248	0.458	0.294	0.477	0.523	0.99

**Table 3 animals-10-00010-t003:** Average measurement results at 8 weeks of life split by genotype, sex, and breed.

Genotype	Sex	Breed	BW 8 Week [g]	std	N	Min [g]	Max [g]
**CC**	F	KIELECKA	3051	352	24	2240	4030
**CC**	F	LANDES	3678	264	29	3340	4610
**CC**	M	KIELECKA	3623	248	3	3340	3800
**CC**	M	LANDES	4032	318	11	3660	4630
**CT**	F	KIELECKA	3043	253	33	2470	3700
**CT**	F	LANDES	3732	237	49	3250	4240
**CT**	M	KIELECKA	3479	270	17	2960	4000
**CT**	M	LANDES	4245	289	19	3720	4780
**TT**	F	KIELECKA	3079	256	24	2610	3750
**TT**	F	LANDES	3665	298	22	3060	4260
**TT**	M	KIELECKA	3440	342	7	2960	4040
**TT**	M	LANDES	4246	495	7	3760	5070

**Table 4 animals-10-00010-t004:** Average measurement results at 12 weeks of life split by genotype, sex, and breed.

Genotype	Sex	Breed	BW 12 Week [g]	std	N	Min [g]	Max [g]
**CC**	F	KIELECKA	3540	285	24	3090	4400
**CC**	F	LANDES	4363	305	29	3930	5510
**CC**	M	KIELECKA	4477	536	3	3960	5030
**CC**	M	LANDES	4989	381	11	4170	5400
**CT**	F	KIELECKA	3463	272	33	2900	4070
**CT**	F	LANDES	4287	717	49	4195	5070
**CT**	M	KIELECKA	4053	357	17	3470	4770
**CT**	M	LANDES	5233	326	19	4760	6010
**TT**	F	KIELECKA	3471	308	24	2480	4000
**TT**	F	LANDES	4405	434	22	3400	5100
**TT**	M	KIELECKA	3843	315	7	3320	4370
**TT**	M	LANDES	5207	747	7	4350	6460

**Table 5 animals-10-00010-t005:** Average measurement results at 95 weeks of life split by genotype, sex, and breed.

Genotype	Sex	Breed	BW 95 Week [g]	std	N	Min [g]	Max [g]
**CC**	F	KIELECKA	4446	289	24	3935	5070
**CC**	F	LANDES	6152	697	29	4260	7945
**CC**	M	KIELECKA	4905	748	3	4160	5655
**CC**	M	LANDES	5776	451	11	5145	6425
**CT**	F	KIELECKA	4581	519	33	3555	5695
**CT**	F	LANDES	6345	625	49	5340	8260
**CT**	M	KIELECKA	4673	405	17	3940	5400
**CT**	M	LANDES	5968	469	19	5275	6870
**TT**	F	KIELECKA	4489	383	25	3890	5265
**TT**	F	LANDES	6191	973	22	4445	9085
**TT**	M	KIELECKA	4659	467	7	4250	5625
**TT**	M	LANDES	6236	722	7	5520	7430

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
