# Peer review of "Polymorphism of the Myostatin (MSTN) Gene in Landes and Kielecka Geese Breeds"

_animals, 2019, doi:10.3390/ani10010010_

Round 1

Reviewer 1 Report

The authors have improved the manuscript with my earlier comments. However, still the paper should be corrected minorly during the copy editing / proof reading, if the editor decide to accept the manuscript. 

Author Response

Dear Reviewer,

Thank you for your comments. All of them were helpful and significantly improved the manuscript.

Manuskript was corrected according to your comments.

Kindly regards,

Response to Reviewer 1 Comments

Point 1. The authors have improved the manuscript with my earlier comments. However, still the paper should be corrected minorly during the copy editing / proof reading, if the editor decide to accept the manuscript. 

Response 1:    The manuscript has been re-checked and all minor corrections have been made.

Reviewer 2 Report

The manuscript has been greatly improved. Just a minor revision is required: Line 129-130 The equation is better to be edited using MathType.

Author Response

Dear Reviewer,

Thank you for your comments. All of them were helpful and significantly improved the manuscript.

Manuskript was corrected according to your comments.

Response to Reviewer 2 Comments

Point 1: The manuscript has been greatly improved. Just a minor revision is required: Line 129-130 The equation is better to be edited using MathType.

Response 1: The equation was edited in MathType 7 and placed in the manuscript.

This manuscript is a resubmission of an earlier submission. The following is a list of the peer review reports and author responses from that submission.

Round 1

Reviewer 1 Report

The manuscript by Smołucha et al., describes polymorphism of the myostatin gene in Landes and Kielecka geese breeds.

Although the information is new to these waterfowl, the manuscript presentation should be greatly improved. The present version makes hard for understanding by this reviewer, and may be by any other reader. Therefore, the manuscript should be completely modified by English as well as science editor/expert. 

The methods section should be modified and presented clearly. For instances, "Animals" section contain sentences that already given in the introduction. Too much description about the body weight measurement. DNA isolation, PCR, sequencing, and alignment should be presented in separate sections with adequate details.

Figure legends 2 to 4 should be simplified. I recommend to include these 1-12 genotype information directly on the figure.  

The authors may also include a phylogenetic tree and sequence alignment (marking exon 1~3) of MSTN CDS sequences from goose, duck, and chicken. Also, mark the location of polymorphism identified in this study on the alignment. Therefore, we can understand the conservation of MSTN gene between these birds, and the polymorphism and associated trait in a comparative manner.  

The discussion should be presented with more details on the polymorphism and associated traits reported in other birds and animals.

Discussion or possible reason of the statement given in the conclusion is needed, especially regarding "only at the 12th week of life". 

Author Response

Dear Reviewer,

Best regards

Reviewer 2 Report

The authors investigated the associations between a mutation in MSTN gene and body weight at different ages in Landes and Kielecka geese. The results of this study provide some useful information. However, the manuscript needs an improvement.

Major comments

Line 121-127: In statistical analyses, the association between the SNP and body weight for two breeds were investigated together. From Part Introduction, we know that the two breeds were largely different from each other. Why not analyze the association separately for each breed? Moreover, does the two populations had their pedigrees recorded? If so, the model can include the random additive genetic effect.

Line 132-133: Was the mutation a synonymous mutation?

Author Response

Dear Reviewer

Best regards
